# Iron Metabolism in Obesity and Metabolic Syndrome

**DOI:** 10.3390/ijms21155529

**Published:** 2020-08-01

**Authors:** Álvaro González-Domínguez, Francisco M. Visiedo-García, Jesús Domínguez-Riscart, Raúl González-Domínguez, Rosa M. Mateos, Alfonso María Lechuga-Sancho

**Affiliations:** 1Inflammation, Nutrition, Metabolism and Oxidative Stress Study Group (INMOX), Biomedical Research and Innovation Institute of Cádiz (INiBICA), Research Unit, Puerta del Mar University Hospital, 11009 Cádiz, Spain; alvaro.gonzalez@inibica.es (Á.G.-D.); franciscom.visiedo@inibica.es (F.M.V.-G.); rosa.mateos@uca.es (R.M.M.); 2Pediatric Endocrinology, Department of Pediatrics, Puerta del Mar University Hospital, 11009 Cádiz, Spain; jesus.dominguezriscart@gmail.com; 3Department of Chemistry, Faculty of Experimental Sciences, University of Huelva, 21007 Huelva, Spain; raul.gonzalez@dqcm.uhu.es; 4Area of Biochemistry and Molecular Biology, Department of Biomedicine, Biotechnology and Public Health, University of Cádiz, 11519 Cádiz, Spain; 5Area of Pediatrics, Department of Child and Mother Health and Radiology, Medical School, University of Cádiz, 11002 Cádiz, Spain

**Keywords:** anemia, childhood obesity, iron, metabolic syndrome, metabolism, obesity

## Abstract

Obesity is an excessive adipose tissue accumulation that may have detrimental effects on health. Particularly, childhood obesity has become one of the main public health problems in the 21st century, since its prevalence has widely increased in recent years. Childhood obesity is intimately related to the development of several comorbidities such as nonalcoholic fatty liver disease, dyslipidemia, type 2 diabetes mellitus, non-congenital cardiovascular disease, chronic inflammation and anemia, among others. Within this tangled interplay between these comorbidities and associated pathological conditions, obesity has been closely linked to important perturbations in iron metabolism. Iron is the second most abundant metal on Earth, but its bioavailability is hampered by its ability to form highly insoluble oxides, with iron deficiency being the most common nutritional disorder. Although every living organism requires iron, it may also cause toxic oxygen damage by generating oxygen free radicals through the Fenton reaction. Thus, iron homeostasis and metabolism must be tightly regulated in humans at every level (i.e., absorption, storage, transport, recycling). Dysregulation of any step involved in iron metabolism may lead to iron deficiencies and, eventually, to the anemic state related to obesity. In this review article, we summarize the existent evidence on the role of the most recently described components of iron metabolism and their alterations in obesity.

## 1. Childhood Obesity

Chronic diseases, also known as non-communicable diseases (NCDs), are the result of a combination of behavioral, environmental, genetic and physiological factors, being responsible for the 75% of deaths in low- and middle-income countries. Among them, obesity, along with cancer, cardiovascular diseases and diabetes, is one of the main NCDs worldwide [1]. Obesity is defined as an excessive fat accumulation that may have detrimental effects on health. Particularly, childhood obesity has become one of the main public health problems in the 21st century, since its prevalence has widely increased in recent years. In 2016, over 41 million children under five years of age were overweight or obese around the world. In developing countries, the prevalence of childhood obesity and overweight children has increased up to 30% of preschool children. If the actual growth rate is sustained, 70 million children will be obese or overweight in 2025. Obese children are more likely to become obese adults, with higher probabilities of suffering other NCDs like diabetes or cardiovascular diseases. Thus, childhood obesity prevention and treatment strategies need to be given primary importance [2,3].

## 2. Metabolic Syndrome

Over 35% of obese children under 10 years of age present at least one cardiovascular risk factor or metabolic disturbance (e.g., arterial hypertension, dyslipidemia, disturbed glucose tolerance), with insulin resistance (IR) being the most common and the earliest to appear. It is already found in patients under five years old [4]. This situation, known as metabolic syndrome (MetS), is a group of cardiovascular risk factors associated with IR of which the main clinical alterations are cellular dysfunction, high fasting triglycerides and glucose, hypertension, low high density lipoprotein cholesterol (HDL-col), systemic inflammation and visceral obesity. It was Gerald Reaven who first named this cluster of risk factors “insulin resistance syndrome” (IRS) after discovering an inverse relationship between MetS and insulin sensitivity, as insulin resistance seems to be the main determinant in the development of this syndrome [5]. In obesity, an increased amount of visceral adipocytes is translated into increased chemo-attractants, meaning higher macrophage infiltration and cytokine release, contributing to systemic inflammation. Likewise, adipocyte dysfunction is accompanied by reduced adiponectin production and higher release of fatty acids. Released free fatty acids may in turn alter the mitochondrial function in peripheral tissues and consequently increase oxidative stress (OS), which lowers the insulin sensitivity of those tissues. Accordingly, insulin production requirements become greater, and the acquired resistance raises circulating glucose levels, increasing the risk of developing type 2 diabetes mellitus (T2DM). There is not yet an international consensus in the definition of childhood MetS; however, it is widely accepted that four components have to be included, namely abdominal obesity, arterial hypertension, dyslipidemia and altered glucose metabolism, so that MetS is diagnosed when abdominal obesity is accompanied by at least two of these other clinical alterations [6,7,8,9].

## 3. Obesity Comorbidities

As the prevalence and severity of obesity increases at young ages, it is common to find obese children with elevated transaminases and nonalcoholic fatty liver disease (NAFLD), which has become the major hepatic disease in children. Thus, NAFLD and MetS are strongly inter-related, and IR is thought to drive the appearance of this “hepatic manifestation of MetS” [6,7,8,9]. Fat deposition in the liver (distinctive for NAFLD) is a consequence of obesity in all ages [10,11], and it may progress to more serious liver dysfunction. Renal pathologies such as hyperfiltration and microalbuminuria may be also associated with obesity [12]. Childhood obesity-associated dyslipidemia increases the risk of suffering it in adulthood [13], and childhood type 2 diabetes mellitus prevalence also increases with obesity [14]. Vitamin D and iron deficiencies have been found to be increased in overweight and obese children, proving that obesity can be considered a quantitative and qualitative form of malnutrition [15,16]. Finally, anxiety, depression, stress and other psychological conditions have all been reported to be more common in obese children and adolescents [10,17].

Childhood obesity is also strongly associated with hypertension, which increases the risk of vascular endothelial dysfunction in adult life [18]. Both leptin and adiponectin, adipocyte-secreted cytokines, have a direct effect on endothelial function along with chemerin, a newly described adipokine involved in the obesity-inflammation cycle; its levels have been described to be higher in obesity [19,20,21]. Non-congenital cardiovascular disease (CVD) is becoming more prevalent in children along with the rise in childhood obesity [22]. Body mass index (BMI), IR and cardiovascular parameters such as left ventricular (LV) mass seem to have a positive relationship in children with obesity [23,24,25]. It is also known that atherosclerosis may start already in children under 10 years of age. In this sense, metabolic parameters in children such as cholesterol, blood pressure, triglyceride concentration or BMI, all them involved in higher risk for cardiovascular disease, have been found to be tightly correlated with adult levels [26].

Obese patients have higher plasma concentrations of all pro-thrombotic factors when compared to non-obese controls [27], possibly due to the hyperinsulinemic state [28]. Obese patients have higher plasma concentrations of PAI-1 (plasminogen activator inhibitor-1), the main inhibitor of plasminogen activation, which is synthetized in adipose tissue and of which the levels decrease after weight reduction [29]. Obese women are also characterized by higher plasma concentrations of anti-thrombotic factors [27], which may be interpreted as a response to the rise in pro-thrombotic factors. However, even when the anti-thrombotic factors’ concentration is higher, their activity is decreased in obese subjects (i.e., t-PA activity) [28]. Obesity is also characterized by resistance to activated protein C, another antithrombotic protein [30]. In diseases like type 2 diabetes, amelioration of insulin resistance and reduction of insulin levels recover the fibrinolytic function [31]. Insulin acts by inhibiting platelet-collagen interactions, reducing aggregation to several agonists [32]. In obese subjects, insulin loses its inhibitory capacity over platelet thrombus formation, facilitating the development of atherothrombotic diseases in IR [32,33].

## 4. Obesity, Inflammation and Oxidative Stress

Obesity is characterized by a low-grade systemic chronic inflammatory state. This chronic inflammation courses with changes in adipose tissue and inflammatory cells (like neutrophils, monocytes, lymphocytes and tissue-specific macrophages) and tissue destruction, leading to an increased level of plasma inflammatory markers and inflammatory cells in circulation [34].

Various mechanisms are involved in this chronic inflammatory state. Lysosomal defective function has been described in obese mice. This dysfunction is accompanied by an induction in the expression of CD36, and CD36 knockout mice are protected from this lysosomal impairment. CD36 mediates the activation of inositol, which in turn increases the calcium influx into the lysosome, leading to its functional impairment [35]. As reviewed by Pessentheiner et al., the role of proteoglycans in the development of metabolic disturbances and low-grade inflammation has acquired increasing relevance. These glycoproteins located in the cell surface interact with a great number of proteins involved in metabolic homeostasis and inflammation [36]. Obesity is also able to cause lymphatic impairment, as inflammatory cells surround the lymphatic vessels and alter anion exchange and lymphatic endothelial cells. This may contribute to low-grade tissue inflammation and the accumulation of inflammatory cytokines [37]. Obesity also mediates changes in bone-marrow, where immune cells are originated. This stem cell differentiation impairment affects systemic inflammatory pathways [38].

In the adipose tissue, an excessive caloric uptake by adipocytes induces the release of numerous inflammatory mediators, such as tumor necrosis factor α (TNFα) and interleukin 6 (IL-6). These in turn stimulate C-reactive protein (CRP) release. On the other hand, adiponectin secretion is reduced, and all these processes result in a pro-inflammatory environment [39]. In addition to the raise in cytokine production, obesity is characterized by increased adhesion molecule levels, which, along with the mentioned cytokines, stimulate tissue-specific macrophage recruitment. Such local inflammation eventually induces IR [40]. Based on the analytical changes in the chemokine, adipokine and cytokine profiles of obese children, Rivera et al. developed a diagnostic tool for IR, rendering a sensitivity and specificity of 93.2%, including levels of leptin, triglyceride:HDL-col ratio, insulin-like growth factor-I (IGF-I), TNFα, monocyte chemoattract protein 1 (MCP1) and pro-fibrotic platelet-derived growth factor (PDGF-BB) [41]. In obese children, higher obesity severity is associated with higher free fatty acid (FFA) levels. IL-6 levels were found to be related to adipose IR [42]. In childhood obesity, adipokines like leptin, IL-6, TNFα and progranulin are increased [43,44]. In addition, leukocyte, lymphocyte, erythrocyte, platelet, CRP and transaminase levels have been found to be higher in obese/overweight children when compared with a control group as a sign of a pro-inflammatory status [45].

OS and inflammatory milieu are known to be interrelated. Whether OS leads to inflammation or if it is the proinflammatory state that leads to a pro-oxidant environment remains controversial. OS seems to play a role in the development of chronic inflammatory diseases. Inflammatory stimuli lead to peroxiredoxin 2 (PRDX2) release. PRDX2, an enzyme with redox capacity acting as an inflammatory mediator, induces macrophages to release TNFα [46]. On the other hand, inflammatory cells release several reactive species at the site of inflammation, leading to OS. Additionally, reactive species may also be released by nonphagocytic cells in response to cytokines. Those reactive species can in turn activate signaling cascades that stimulate proinflammatory gene expression. In this line, hydrogen peroxide (H_2_O_2_) may lead to inflammation through NF-κB activation. OS is also involved in the activation of NOD-like receptor protein 3 (NLRP3) inflammasome, a complex that triggers innate immune defenses by maturation of proinflammatory cytokines [47].

OS also plays a role in the development of MetS. Levels of HDL-col, well known for its antioxidant and anti-inflammatory actions, are lower in obese adults and children with MetS. Recent studies also link gamma-glutamyl transferase (GGT) to MetS, given its relation to glutathione metabolism and thus antioxidant defense [48]. Our group previously described marked differences in oxidative stress biomarkers and antioxidant response between obese children and controls, and even between metabolically healthy and metabolically unhealthy obese children. Obese children with insulin resistance presented higher levels of thiobarbituric acid reactive substances (TBARS) as a lipid peroxidation marker after an oral glucose tolerance test (OGTT) when compared to controls and obese children without insulin resistance. They also presented an impaired capacity to activate catalase activity after an OGTT [49]. Along with these results, Kilic et al. found that obese children presented an increase in both total oxidant and antioxidant status [50]. Isoprostane (lipid peroxidation marker) is also higher in obese children and is related to urinary H_2_O_2_, high-sensitivity CRP, HOMA-IR (HOmeostatic Model Assessment for Insulin Resistance) and triglycerides [51]. In prepubescent children, total antioxidant capacity is inversely related to the percentage of fat mass and waist circumference [52]. Of note, lower levels of serum antioxidants of nutritional origin, (such as retinol, β-carotene and vitamin E) are related to increased metabolic alterations in obese adolescents [53].

## 5. Childhood Obesity and Anemia

In 1962, Wenzel et al. were the first to find lower serum iron levels in obese patients when compared with a control group [54]. In the past decades, numerous theories have been raised to attempt to give an explanation of this fact, such as dietary deficiencies of iron (which has not been reproduced later [55]), higher iron requirements due to a higher blood volume, or lower myoglobin in the muscle due to the absence of physical activity [56]. However, in the past few years, the chronic inflammatory state associated with obesity and the role of hepcidin have gained great relevance.

The main physiological characteristic of obesity is the increased number of adipocytes and their greater size, but adipocytes also suffer other functional changes of great importance. In obesity, as mentioned above, there is constant macrophage infiltration into adipose tissue and changes in the local production of pro-inflammatory cytokines such as interleukin 1 and 6 (IL-1, IL-6) and tumor necrosis factor α (TNFα) [57]. These changes in cytokine production could lead to impaired erythropoietin production and altered response of erythroid precursors, a recognized mechanism of anemia associated with chronic diseases [58].

Zhao et al. found lower serum iron concentrations and transferrin saturation in a group of obese/overweight participants when compared to a control group [59]. Manios et al. found ferritin levels to be higher in a group of obese school children aged between 9 and 13 when compared with a control group, together with a negative correlation between transferrin saturation and adiposity [60]. Del Giudice et al. proved iron levels and transferrin saturation to be lower in obese children, whereas hepcidin levels where higher in the obese group when compared with control participants [61]. A study performed in 2018 found higher levels of C reactive protein (CRP), hepcidin, leucocytes, platelets, leptin and total iron binding capacity (TIBC) in an obese population when compared with controls. On the other hand, mean corpuscular volume (MCV), adiponectin levels and transferrin saturation were lower in the obese group. They did not find any differences in hemoglobin, serum ferritin, iron and IL-6 [62]. However, a third of MetS patients suffer from hyperferritinemia with a normal transferrin saturation. This correlation is called “dysmetabolic iron overload syndrome” (DIOS) [63]. Bertinato et al. found diet-induced obese rats to have lower mean corpuscular hemoglobin (MCH) and liver iron concentrations [64]. As an indirect marker of bone marrow turnover and anemic state our group found volumetric dispersion of erythrocytes to be higher in a group of prepubescent children (Figure 1). Supporting our finding, Fujita et al. found red cell distribution width (RDW) to be higher in obese and overweight adolescents when compared with a control group [65]. With a group of 110 children aged between 6 and 16 years (50 obese and 60 healthy children), Doğan et al. also found that RDW and ferritin levels were significantly higher in obese children when compared to the control group, while having lower serum iron and transferrin saturation levels [66].

## 6. Iron Metabolism

Iron is the second most abundant metal on Earth, being crucial for almost every living organism. However, its ability to form highly insoluble oxides when it is in contact with oxygen reduces its bioavailability. Moreover, according to the World Health Organization (WHO), iron deficiency is the most common nutritional disorder in the world [67]. To counteract this, humans and all living organisms have developed efficient mechanisms to capture iron in useful states [68]. Humans need iron as a cofactor for hemoproteins (catalase (CAT), cytochromes, hemoglobin or myoglobin) and other non-heme proteins involved in cell proliferation and differentiation, DNA synthesis or drug metabolism [69]. On the other hand, iron can also lead to the formation of toxic oxygen free radicals (hydroxyl radical) through the Fenton reaction. Thus, iron metabolism needs to be tightly regulated at cellular and organism levels. Iron regulation is strictly dependent on redox state (Fe^2+^/Fe^3+^); to be absorbed, and when serving as a cofactor, it needs to be in the reduced state (Fe^2+^), but when transported by transferrin or stored as ferritin, it is in its oxidized form (Fe^3+^). At the cellular level, iron regulation is accomplished by iron importers, iron storage proteins or iron efflux pumps among others, which allow the transport of iron to its sites of utilization, minimizing its availability for reactive oxygen species (ROS) generation. Once in the cell, iron is submitted to compartmentalization: mitochondria for heme biosynthesis, iron-containing and iron regulatory proteins in the cytoplasm, etc. At the whole organism level, iron metabolism is regulated at different levels: absorption (by enterocytes in the duodenum), utilization (mainly erythrocyte production in bone marrow), storage (liver) and recycling (spleen and reticuloendothelial system, RES) [70]. We summarize physiological iron metabolism in Figure 2 and proceed to detail every step of it.

### 6.1. Iron Absorption

In humans, the absorbed iron is limited to approximately 10% of the total iron consumed through the diet. This absorption takes place mainly in the duodenum by the action of enterocytes, which absorb heme-bound iron via the heme carrier protein 1 transporter (HCP1), also known as PCFT/HCP1, since this protein is a proton-coupled folate transporter (PCFT) [69,71]. In the enterocyte, heme can be degraded or absorbed into the circulation. Heme-oxygenase (HO-1), in association with cytochrome P450 (CYP450), is the enzyme responsible for the degradation of absorbed heme groups in the enterocyte to release free iron (Fe^2+^), which will join the labile iron pool (LIP), and generate carbon monoxide and biliverdin, subsequently converted into bilirubin by biliverdin reductase [72,73]. Alternatively, intact heme absorption into the circulation is carried out by two transporters in the basolateral membrane of the enterocyte: the breast cancer-resistant protein (BCRP) and the feline leukaemia virus subgroup C (FLVCR) [74].

The most common non-heme iron species found in food is the ferric form (Fe^3+^) bound to citrate (or acetate), a highly insoluble and non-easily absorbable form. To accomplish its absorption, ferric iron is reduced to ferrous iron (Fe^2+^) by means of the low pH in the stomach or the presence of other metabolites like dietary ascorbic acid (vitamin C). In addition to these two, the enterocyte has its own mechanisms to reduce Fe^3+^ to allow its absorption. Indeed, at the enterocytes’ apical membrane, facing the gut lumen, there are two proteins capable of the enzymatic reduction of ferric iron using electrons from the oxidation of ascorbic acid (known as ferrireductases): the duodenal cytochrome b (Dcytb) and the six-transmembrane epithelial antigen of the prostate 2 (STEAP2). Once reduced, divalent iron can be transported into the enterocytes through the divalent metal transporter 1 (DMT1, also called Nramp2, SLC11A2 and DCT1) or the zinc transporter Zrt–Irt-like protein 14 and 8 (ZIP 14/8). Protons, needed as the driving force for iron transport, are supplied by a sodium/hydrogen exchanger (NHE3) [75,76,77].

Finally, enterocytes are also able to absorb dietary ferritin by an endocytosis mechanism related to the adaptor-related 2 protein complex (AP2) [78].

### 6.2. Iron Storage

Once in the enterocyte, ferrous iron may either be stored or sent to the circulation to be transported by the liver-delivered protein transferrin (Tf). In the enterocyte, ferrous iron becomes part of the labile iron pool (LIP). The rest is transported to form different cellular stores in the form of ferritin. The molecule responsible for iron incorporation into ferritin is the poly (rC) binding protein 1 (PCBP) chaperone. In fact, PCBP chaperones also transfer ferrous iron to other proteins that are able to bind iron as a cofactor, such as myoglobin, CAT and cytochromes [77]. Ferritin is composed of two different subunits known as heavy (H) and light (L) chains, which form a space with a diameter of 8 nm, in which 4500 iron ions (Fe^3+^) can be stored. To that end, H-chains have ferroxidase activity for the oxidation of ferrous to ferric iron, and acidic residues contained in L-chains are crucial for the nucleation of the oxidized ferric iron within the formed core [46]. Ferritin can also be found in plasma, and two different mechanisms have been described for ferritin uptake by the cell, one mediated by an H-ferritin receptor called TIM2 (T-cell immunoglobulin and mucin domain-containing protein 2) and the second one mediated by the L-ferritin receptor Scara5 (scavenger receptor class A, member 5) [15,79].

### 6.3. Iron Transport and Utilization

Ferroportin 1 (FPN1, also called MTP1, IREG1 and SLC40A1) regulates the transport of ferrous iron across the enterocyte’s basolateral membrane into the blood stream [80]. FPN1 is a cytoplasmic protein that needs to be translocated to the cell membrane by mon1a [81]. Ferrous iron now must be oxidized back to the ferric form to be transported by transferrin (Tf). To accomplish this, hephaestin (HEPH), a multi-copper ferroxidase enzyme in the enterocyte’s basolateral membrane coupled to FPN1, catalyzes the oxidation of ferrous iron to ferric iron. Ceruloplasmin (CP) also acts as a ferroxidase enzyme in some tissues. In fact, deletion of HEPH and CP genes leads to systemic iron deficiency [82,83]. Once oxidized, Fe^3+^ binds to Tf, which has two iron-binding sites. When Tf is iron-free it is called Apo-Tf, and if Tf is saturated with two ferric iron atoms it is called Holo-Tf. Transferrin receptor (TfR) affinity for Tf depends on this saturation state, being much greater for Holo-Tf than for Apo-Tf [84]. Ferritin degradation has been recently proposed as an alternative mechanism for iron release. Such degradation takes place by lysosome and proteasome-mediated mechanisms. In this sense, nuclear receptor coactivator 4 (NCOA4) seems to act as the receptor binding ferritin and delivering it to lysosomes [85].

TfR1, expressed in almost every nucleated cell, and in a lower proportion in DMT1 and ZIP14, is implicated in transferrin-bound ferric iron (TBI) absorption [77]. As mentioned above, TfR1 has a high affinity for TBI. Binding of hemochromatosis protein (HFE) to TfR1 decreases the affinity between TfR1 and holo-Tf. Subsequently, endocytosis of the TBI-TfR1 takes place and clathrin-coated endosomes are formed. Once clathrin is removed, ATP-dependent proton pumps acidify the endosome matrix and, inducing conformational changes in Tf and TfR1, dissociate ferric iron from the complex. However, Tf remains bound to its receptor until it is again transferred to the cell membrane by the trafficking protein Sec15l1. Once outside the cell, external pH leads to the release of Apo-Tf. Dissociated iron may then be reduced to Fe^2+^ by another ferroreductase, the six-transmembrane epithelial antigen of the prostate 3 (STEAP3). Finally, ferrous iron may exit the endosome via DMT1 to be used/stored by the cell [84,86].

As well as TfR1, there is another isoform of the transferrin receptor: TfR2. TfR2 is mainly expressed in hepatocytes and erythroid precursors, and its main function seems to be as an iron sensor, since holo-Tf upregulates its expression. In erythroid precursors, TfR2 plays a relevant role in erythropoiesis by biding to the erythropoietin receptor and facilitating ferric iron transport from lysosomes to mitochondria safely, both in erythroblasts and dopaminergic neurons [86]. Moreover, there is another member of the Tf family first described in human milk and named lactoferrin (hLf), with lower affinity to ferric iron than transferrin. Lactoferrin is thought to be involved in the defense against inflammation processes [87].

Besides the transferrin, there is a transferrin-independent iron influx/efflux system: the neutrophil gelatinase-associated lipocalin (NGAL, LCN2)–24p3R/megalin–siderophore axis. NGAL, part of the lipocalin family, binds to siderophores, which are high-affinity iron chelators. The complex formed by NGAL, ferric iron and a siderophore binds to the cell-surface receptor 24p3R (also known as SLC22A17) or megalin and is susceptible to receptor-mediated endocytosis. This 24p3R/megalin cell-surface receptor also mediates the uptake of apo-NGAL, so it can bind ferric iron within the cell and make it exportable [88,89].

In mitochondria, after mitoferrin-mediated ferrous iron transport through the inner membrane, frataxin (FXN) seems to be responsible for delivering iron to a ferrochelatase involved in the final step of heme production, the insertion of ferrous iron into protoporphyrin IX (PPIX). By the inhibition of delta aminolevulinate synthase (ALAS), a catalyst of the first step in porphyrin ring production, regulated by hemin (the oxidized form of heme), or induction of HO1, heme serves as a regulator of its own homeostasis. Iron is also used in the synthesis of iron-sulfur clusters (ISCs) for later incorporation into electron transfer proteins. FXN is also one of the components of a multimeric protein core complex named ISCU, involved in the production of those ISCs. The formed cluster can either be used in the mitochondria or transported to the cytoplasm through the ABC transporter ABCB7. Recently, FXN has been described to act as a main regulator of ferroptosis, (a recently described novel form of cell death), by regulating mitochondrial function and iron homeostasis [70,90,91,92,93].

### 6.4. Iron Recycling

Humans lack an active system of iron secretion, but about 1–2 mg of iron is lost by means of skin desquamation, bleeding, infestations, etc. The RES is responsible of the recycling of about 25 mg of iron every day, pointing to the great relevance that iron recycling displays in iron homeostasis in humans [94].

To maintain the required rate of erythropoiesis (200 billion new erythrocytes every day) and the subsequent heme synthesis, 24 mg of iron are needed per day. To cover these iron requirements, macrophages are deeply involved in iron recycling in various ways. Macrophages present different iron uptake mechanisms such as the usual import of transferrin-bound or non-transferrin-bound iron, ferrous iron uptake via natural resistance-associated macrophage protein 1 (NRAMP1) or Dmt1 (also known as NRAMP2, as mentioned above) or senescent erythrocyte phagocytosis [95,96]. To accomplish phagocytosis, macrophages recognize some ageing signals exhibited by senescent erythrocytes, such as an increase of membrane phosphatidylserine and a decrease in membrane flexibility or the antigen CD47 [97]. Erythrocytes are then incorporated into the denominated phagolysosomes, where they are degraded by ROS and hydrolytic enzymes until heme release. Released heme is degraded by the above-mentioned HO-1 into biliverdin, carbon monoxide and ferrous iron. Iron transport within the macrophage is performed in three stages. First, heme is transported to the cytosol across phagosome membrane by heme responsive gene 1 (HRG1). After its degradation, ferrous iron is transported through the cytoplasm by PCBP1, and is finally delivered to FPN1 to be exported and reduced by CP in circulation, making possible its transport through the organism via Tf [76,95,96].

With hemolysis, hemoglobin is released into the circulation. Hemoglobin in plasma is bound by haptoglobin, and the hemoglobin-haptoglobin complex is recognized by the CD163 receptor in monocytes and macrophages, following its endocytosis and degradation inside the macrophages. If the hemolysis rate is high and haptoglobin is saturated, hemoglobin is degraded into heme, and the free heme groups bind to hemopexin and again the complex is recognized and phagocytosed by macrophages via the low-density lipoprotein receptor related protein LRP/CD91 [95,96].

As discussed earlier, ferrous iron may be drawn back to the circulation thanks to FPN1 and, after being oxidized by ceruloplasmin, join apo-Tf and enter the cycle again. Macrophage cytoplasmic ferrous iron may also be stored in ferritin. After suffering a lysosomal partial digestion, ferritin may lead to the formation of hemosiderin, an insoluble iron store primarily found in macrophages [95,96].

### 6.5. Iron Homeostasis Regulation

The antimicrobial hepcidin preprotein is produced in the liver as a product of the expression of the *HAMP* gene. To achieve its functional conformation, hepcidin must undergo proteolytic cleavage by the enzyme furin, leading to the production of the active peptide hormone composed by 25 amino acids. Hepcidin effects tend towards the reduction of circulating iron levels acting at two levels. In cells like macrophages or hepatocytes, hepcidin binds to ferroportin and leads to its internalization and degradation, so that less ferrous iron is exported to circulation. In enterocytes, hepcidin seems to inhibit DMT1 production transcriptionally, inducing a decrease in ferrous iron uptake in the duodenum [98,99].

Hepcidin levels are in turn controlled by numerous hepatic proteins. Hemojuvelin (HJV) is a membrane protein with a pivotal role in the control of hepcidin expression. In the membrane, HJV acts as a co-receptor and mediates the binding between the bone morphogenic protein receptor (BMPR) and its substrate, the bone morphogenic protein (BMP). This complex leads to the subsequent phosphorylation of cytosolic small-mothers-against-decapentaplegic proteins (SMADs). Then, a complex of SMAD proteins enters the nucleus and activates the transcription of the *HAMP* gene after binding to the BMP responsive element at its promoter. After being cleaved by matriptase-2, HJV is converted into a soluble form that may act as an antagonist in the interaction BMP-BMPR, inhibiting hepcidin expression [99,100,101]. The hemochromatosis protein (HFE) and both transferrin receptors (TfR1 and TfR2) are also part of a hepcidin level regulatory pathway via the ERK/MAPK pathway. With high plasma iron levels, as TfR1 presents a higher affinity for holo-Tf than for HFE, HFE dissociates and binds to TfR2. Thereafter, the holo-Tf-HFE-TfR2 complex activates the ERK signaling pathway, resulting in *HAMP* expression [99,101]. Erythropoiesis also inhibits hepcidin synthesis in hepatocytes, because erythroid precursors release erythroferrone (ERFE), which suppress hepcidin expression to facilitate iron acquisition for hemoglobin synthesis [102,103].

As the reduction of iron levels limits the growth of iron-dependent micro-organisms, hepcidin expression is a defensive mechanism after infection and inflammation. Increased levels of cytokines such as IL-6 after infection lead to the activation of Janus kinases (JAK), which phosphorylate signal transducers and activators of transcription (STAT) proteins. STAT3 can now enter the nucleus and activate *HAMP* expression [99,104].

Another mechanism of iron homeostasis regulation is displayed by iron regulatory proteins (IRP) and iron responsive elements (IRE). IREs are located in the untranslated region (UTR) of mRNAs that encode iron metabolism proteins. If located in the 5’ UTR, after binding of the IRP, the IRE blocks the translation of the encoded protein. On the other hand, when located in the 3’ UTR, the IRE mediates the stabilization of the mRNA and thus facilitates translation [105].

Finally, hypoxia inducible factor (HIF) is crucial in adaptive responses to low oxygen levels and, at the same time, is regulated by iron and regulates iron homeostasis. Under normal oxygen level conditions, the alpha subunit of HIF is degraded by means of members of the prolyl hydroxylase domain (PHD) family. As PHD activity depends on iron and oxygen levels, under low oxygen conditions PHD activity decreases and HIF is not degraded. Here we discuss the role of the main HIF isoforms involved in iron metabolism regulation, HIF-1 and HI-2. HIF-2 regulates erythropoiesis by affecting erythropoietin hormone levels (EPO) and iron mobilization via the activation of the transcription of DMT1, DcytB and FPN. HIF-2 also represses hepcidin production in the liver. On the other hand, HIF-1 has been proven to regulate TfR1 and HO-1 expression, establishing a link between iron metabolism and hypoxia [106].

Recently, research has focused in the role of microRNAs (miRNAs), small non-coding RNAs, in the regulation of proteins involved in iron homeostasis at every level: iron uptake (TfR and DMT1), iron export (FPN), iron storage (ferritin), iron regulation (HFE and HJV) and even iron utilization (ISCU) [73]. In this line, miRNA122 has been found to act by silencing genes that control systemic iron levels such as *HFE*, *HV*, *BMPR1A* or even *HAMP*. Moreover, iron overload reduces the expression of miRNA122 [107].

## 7. Iron Metabolism in Obesity

Obesity has been found to influence every step of iron metabolism. Furthermore, the implication of each of these steps in the pathophysiology of obesity-related anemia has been studied in animal models, in adulthood obesity and in childhood obesity. In the next paragraphs we summarize the published evidence on this relationship between obesity and iron metabolism from a molecular and mechanistic point of view (Table 1). The described changes are summarized in Figure 3.

### 7.1. Iron Absorption

A study including metabolically healthy and insulin-resistant obese adults found HO-1 levels to be a positive predictor of metabolic disease in humans. Moreover, depletion of HO-1 in mice protects the animal from developing insulin resistance and inflammation related with obesity [108]. In accordance with these results, obese children in Mexico have been found to have increased HO-1 levels [109]. Nonetheless, after treating cultured adipocytes and macrophages with hemin (HO-1 inductor), macrophages experience a switch to their anti-inflammatory phenotype, acting as a defense against obesity-induced inflammation and insulin resistance [110]. Likewise, when an animal model of obesity (mice receiving a high fat diet, HFD) is treated with an HO-1 inducer, it results in improved endothelial cell function and repressed adipogenesis [111].

In obese individuals, a loss of JAK3 mediated phosphorylation of BCRP leads to intestinal dysfunction of this heme transporter [112]. FLVCR1 mRNA expression is positively correlated with fasting glucose and negatively correlated with insulin sensitivity, but no significant changes were observed related to weight gain in HFD-induced obese mice [113].

Although hepcidin implication in obesity-related iron deficiency will be later discussed, Sonnweber et al. described this situation to be derived from impaired ferrous iron absorption in the duodenum. They found Dcytb and hephaestatin oxidoreductase mRNA levels to be lower in HFD-induced obese mice, but found higher levels of TfR1 and DMT1 mRNA and proteins. Iron supplementation reversed the increased TfR1 and DMT1 protein/mRNA levels [114].

After consumption of an HFD for 12 weeks, obese rats exhibited a reduction in the renal expression of the anion exchanger NHE3 [115].

Zinc transporter expression has also been related to obesity in a group of obese Korean women. mRNA levels of many zinc transporters such as ZnT4, ZnT5, ZnT9, ZIPp1, ZIP4 and ZIP6 were significantly lower in obese women [116]. Related to that fact, plasmatic and erythroid zinc levels appeared to be significantly lower in obese children when compared to a control group, and urinary zinc secretion appeared higher, which may have some effect on insulin sensitivity, since zinc is involved in insulin secretion and action [117,118].

### 7.2. Iron Storage

Rodent models lacking the active ferritin subunit (H-Ft) are unable to develop HFD-induced obesity, as the reduction of intracellular iron deposits ameliorates the inflammatory state [119]. In a study on the association of metabolic health in obesity and iron status markers in prepubescent children, Suárez-Ortegón et al. found circulating ferritin levels to be related to metabolically unhealthy obesity [120].

### 7.3. Iron Transport and Utilization

As mentioned above, HFD-induced obese mice present lower hephaestatin mRNA levels [114]. However, adipose tissue from obese humans (both adults and children) seems to contribute to higher circulating CP levels [121,122,123]. Additionally, livers of HFD-induced obese mice presented lower iron levels and increased endoplasmic reticulum stress (ERS), together with an impaired expression of NCOA4 and ferritin [124].

Low concentrations of serum sTfR are indicators of a healthy tissue iron status. In accordance with this, circulating levels of sTfR showed a direct correlation with BMI in 75 hyperferritinemic men [125]. Moreover, elevated sTfR serum levels served as a predictor of a higher risk of developing T2DM in obese individuals [126].

On the other hand, Moreno-Navarrete et al. found circulating lactoferrin concentration to be inversely associated with body mass index in adult men (BMI) [127]. Moreover, the same group also found circulating lactoferrin to be negatively associated with hyperglycemia, inflammatory markers and obesity, and directly associated with insulin sensitivity [128].

LCN2 mRNA and protein levels were higher in obese patients when compared with lean subjects. This increased expression was associated with increased inflammatory markers [129]. Moreover, LCN2 seems to play obesogenic and anti-thermogenic effects. These effects seem to be via the inhibition of BAT activity, since after feeding wild-type and *LCN2* knockout mice with HFD the second group exhibited a lower weight gain and higher BAT activity [130]. However, LCN2 behavior in children seems to be the opposite of that described in adults and in animals, since LCN2 plasmatic levels were found to be decreased in 80 obese girls [131]. Megalin has been shown to play a protective role against HFD-induced obesity in endothelial megalin-deficient mice [132] but data on humans have not been yet published, to the best of our knowledge.

Deletion of FXN in mice leads to an impaired oxidative metabolism accompanied by a higher predisposition of these animals to suffer from high-caloric diet-induced obesity [133], but again, to date, this has not been explored in human obesity.

### 7.4. Iron Recycling

Serum haptoglobin levels were directly related to BMI, HOMA-IR, fasting insulin and blood glucose levels in a group of obese women, and serve as a marker of obesity [134,135]. One possible mechanism for such increased haptoglobin production in obesity is a TNFα-mediated induction, since mice overexpressing TNFα present higher levels of haptoglobin and obese mice lacking this cytokine in white adipose tissue (WAT) resulted in a downregulation of adipose haptoglobin [136]. More recently, in Mexican adults and children, haptoglobin levels were found to be positively associated with obesity. In the same study, the authors found the *HP rs2000999 G* allele to be related to haptoglobin levels, but not obesity [137]. Moreover, the haptoglobin phenotype has been related to oxidative stress in obesity, since the presence of the *Hp 2* allele seems to be associated to lower levels of reduced glutathione (tGSH) in obese children. This difference is even higher if the Hp phenotype is accompanied by *H63D* polymorphism in the *HFE* gene [138]. CD163 expression is also elevated in obese subjects and positively correlates with insulin resistance, as measured by HOMA-IR [139]. Obese children also present higher CD163 levels, associated with markers of liver injury and with metabolic parameters. Treating these children with changes in lifestyle resulted in changes in CD163, associated with, among others, improvement of insulin sensitivity [140].

It is known that hemopexin mRNA levels are increased in inflammatory states, since cytokines such as IL6 or TNFα induce its expression. However, insulin is able to attenuate this overexpression [141]. In adipocyte cell cultures, hemopexin increases with adipogenesis and if hemopexin expression is truncated, adipocyte differentiation is impaired. In humans, hemopexin expression varies according to the metabolic disease status, and serum hemopexin is related to triglyceride levels [142]. Concerning its receptor, exome sequencing of the *LRP1B* gene in a cohort of children suffering from severe obesity revealed a single nucleotide polymorphism (SNP), rs431809, in intron 4 to be significantly related to BMI. Epigenetic modifications in this same intron were also found to be associated with BMI [143].

Obesity leads to a reduction in the iron content of adipose tissue infiltrated macrophages, together with a reduction in the expression of some iron importers, and the level of the main iron exporter, FPN1, was also found to be decreased in the liver of HFD-induced obese mice. Rats with fatty diet induced-obesity present hemosiderin deposits and macrophages filled with hemosiderin droplets [144,145].

### 7.5. Iron Homeostasis Regulation

A prospective analysis in a Chinese cohort has shown a relation between furin levels and the risk of developing abdominal obesity. A population of 892 Chinese adults without abdominal obesity at baseline was followed-up for four years. They found baseline serum furin deficiency to be a contributor to abdominal obesity [146]. Gajewska et al. determined the iron status in a group of 80 children and found the obese ones to have a 40% increase in hepcidin levels, as well as a 30% reduction in FPN1 levels. However, soluble transferrin receptor (sTfR), ferritin, iron or hemoglobin levels were similar, as well as the mean corpuscular volume (MCV), a sign of an iron deficiency-independent alteration of the FPN-hepcidin axis [147]. In accordance, Park et al. found HAMP mRNA to be lower in an HFD-induced murine model of obesity [148].

HJV mRNA expression has been found to be increased in obese patients’ adipose tissue, along with hepcidin mRNA. This results in increased blood soluble HJV [149]. As reviewed by Blázquez-Medela et al., many BMP isoforms are related to obesity and its comorbidities. Higher levels of BMP2 have been described in WAT of overweight and obese individuals when compared to controls. In mice, deletion of BMP4 results in reduced insulin sensitivity and increased adiposity. In animal models of obesity, treatment with intraperitoneal injections of BMP7 resulted in body weight and inflammation reduction. Other BMP isoforms such as BMP8B, BMP3 or BMP3B have also been related to obesity [150]. When analyzing BMPR1A expression in 297 subjects, Böttcher et al. found a direct relationship between mRNA levels in adipose tissue and obesity. They also described single nucleotide polymorphisms (SNPs) likely to be involved in this increased BMPR1A expression [151]. Additionally, HJV repressor matriptase-2 deficient mice are protected from HFD-induced obesity, showing decreased BMI and improved glucose tolerance and insulin sensitivity [152]. Thus, every component involved in the main regulatory pathway controlling hepcidin synthesis is exacerbated in obesity, including the enzyme involved in the cleavage of the active form (furin). To study the implication of SMAD proteins in obesity-related metabolism, Seong et al. developed HFD-induced obese mice overexpressing SMAD isoforms. They found SMAD2, 3 and 4 to improve obesity-related metabolic parameters and inflammation; however, SMAD7 had detrimental effects by regulating MPK38 activity [153].

EPO supplementation in rodent models of obesity decreases body weight gain and glycated hemoglobin levels (HbA1c) [154]. When the expression of the erythropoietin receptor (EPOR) is restricted to hematopoietic tissues and absent in other tissues, mice develop obesity and insulin resistance, together with a lower energy expenditure and greater white fat mass [155]. On the contrary, cells treated with an EPO-derived peptide without erythroid activity but retaining other functions suppress adipogenesis and ameliorate macrophage inflammatory activation. In HFD-induced obese mice these peptides improved obesity and insulin resistance [156], pointing out that the obesogenic and insulin-resistant effect of EPO is intimately related to its hematopoietic effect.

Although increased levels of cytokines such as IL6 are expected to increase hepcidin production via the JAK-STAT pathway, adipose tissue STAT3 mRNA levels have been described to be lower in HFD-induced obese rats and in obese children with hypertriglyceridemia [157]. Even so, in obesity, chronic activation of JAK-STAT3 leads to leptin and insulin resistance in the central nervous system and peripheral organs, respectively [158].

Soluble factors released from obese adipocytes lead to liver HIF1α overexpression and deletion of hepatic HIF1α protection from obesity-induced glucose intolerance, without altering BMI or insulin resistance in murine models [159]. The overexpression of the HIF1 alpha subunit leads to obesity by inhibiting thermogenesis and cellular metabolism in brown adipose tissue (BAT) [160]. This remains controversial, since inhibition of PHD in a murine model of obesity results in the stabilization of HIF1α and, as a consequence, in a reduction of BMI and HDL-col level, as well as the improvement of some obesity-related alterations such as macrophage infiltration into WAT or adipocyte fibrosis [161]. On the other hand, the HIF2 isoform is proposed to have the opposite effect on obesity and its complications than HIF1. Thus, HIF-2α knockout in mice worsens HFD-induced obesity and insulin resistance [162].

miRNAs involved in inflammation and iron metabolism have also been studied in obesity. A study in 2018 found miRNA 155 (related to inflammation) and miRNA 122 (related to iron metabolism) to be increased at both systemic and sperm levels in obese men [163].

## 8. Concluding Remarks

In the last decades, obesity, and in particular childhood obesity, has become a worldwide pandemic, affecting a third of preschool children in developing countries. Both inflammatory and oxidative states have been related to the development of obesity-associated complications. As pediatric obesity increases the risk of developing several comorbidities in both childhood and adulthood, it has become crucial to gather the existing knowledge on the molecular and cellular alterations found in these conditions. Iron deficiency, the most common nutritional disorder in the world, is one of the comorbidities associated with obesity, and although some hypotheses have been suggested, the underlying mechanisms for obesity-associated iron deficiency and anemia remain unclear. Recent evidence shows that iron metabolism alteration might not just be a consequence of obesity, but may play a pivotal role in the development of obesity metabolic derangements.

In this review we attempted to provide an overview of the molecular and cellular perturbations found in obesity and metabolic syndrome related to iron homeostasis and regulation, and how these interact with each other. Specifically, we have revised the evidence of the alterations found in iron absorption, storage, transport, utilization, recycling and homeostasis regulation. We describe how these relate to inflammation and oxidative stress in obesity.

## 9. Future Directions

The complexity of obesity-related iron metabolism alterations reviewed in this work illustrates the need for further research to accomplish future preventive and therapeutic interventions. Less is known on the role of OS in iron metabolism disturbances and its effects at the cellular level. As previously described by our group, metabolic derangements in obese children are related, at least, to higher levels of lipid oxidation markers and a deficient capacity of those children to activate their antioxidant defenses. This increased OS could lead to metabolic dysregulation at several levels and even to cellular damage and destruction of erythrocytes. The question of whether reestablishing iron homeostasis may improve metabolic, inflammatory and OS states could open a completely innovative approach in the treatment of obesity metabolic comorbidities, and remains to be tested.

## Figures and Tables

**Figure 1 ijms-21-05529-f001:**
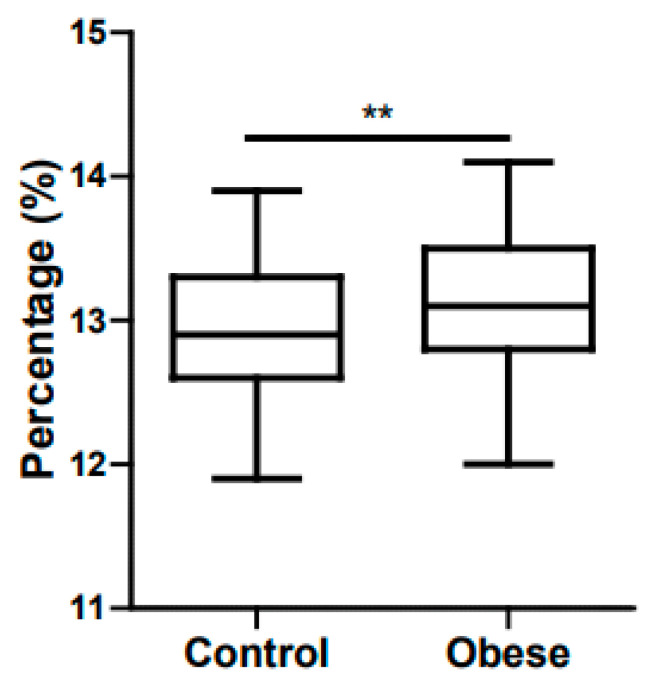
Volumetric dispersion of erythrocytes in pubescent controls versus obese children. ** p < 0.01.

**Figure 2 ijms-21-05529-f002:**
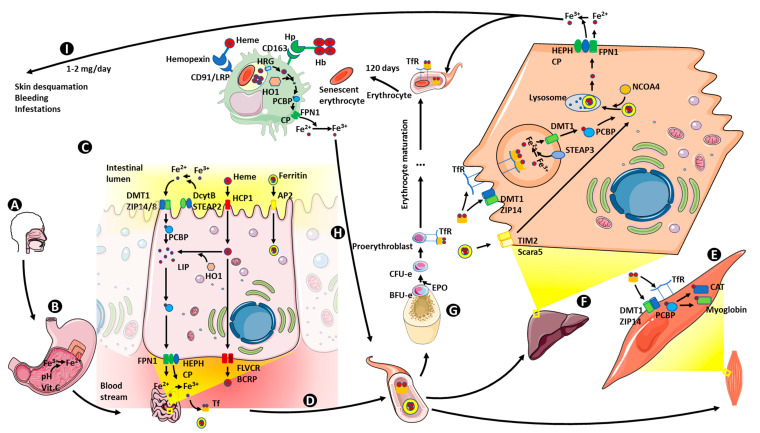
Physiology of Iron Metabolism. Iron ingested from the diet (**A**), is reduced from Fe^3+^ to Fe^2+^ in the stomach (**B**). In the duodenum, enterocytes transport Fe^2+^, heme groups and ferritin across the microvillus membrane (**C**). Fe^2+^ is transported by ferroportin across the basolateral membrane into the portal system and must be oxidized to Fe^3+^ for binding to transferrin and other molecules with high affinity for Fe^3+^ to be transported to the liver. Transferrin-bound iron is necessary for cells expressing transferrin receptors for uptake of iron, mainly for production of heme proteins (**D**). Transferrin-bound iron is taken up by myocytes, where Fe^3+^ is oxidized again to Fe^2+^ in order to be incorporated into myoglobin (**E**), hepatocytes being the main ferritin store (**F**), and by proerythroblasts for synthesis of hemoglobin (**G**). When mature erythrocytes die, macrophages liberate Fe^2+^ from hemoglobin, which is oxidized again to recirculate bound to transferrin (**H**). Finally, 1–2 mg iron is lost per day from the organism by desquamation, bleeding and other mechanisms (**I**). Vit. C: vitamin C; DMT1: divalent metal transporter 1; ZIP 14/8: Zrt–Irt-like protein 14 and 8; DcytB: duodenal cytochrome B; STEAP 2: six-transmembrane epithelial antigen of the prostate 2; HCP1: heme carrier protein 1; AP2: adaptor-related 2 protein; PCBP: poly (rC) binding protein; LIP: labile iron pool; HO1: heme oxygenase 1; FPN1: ferroportin 1; HEPH: hephaestin; CP: ceruloplasmin; FLVCR: feline leukemia virus subgroup C; BCRP: breast cancer-resistant protein; Tf: transferrin; TfR: transferrin receptor; CAT: catalase; Scara 5: scavenger receptor class A, member 5; TIM2: T-cell immunoglobulin and mucin domain-containing protein 2; STEAP 3: six-transmembrane epithelial antigen of the prostate 3; NCOA4: nuclear receptor coactivator 4; BFU-e: burst forming unit-erythroid; CFU-e: colony forming unit-erythroid; EPO: erythropoietin; Hb: hemoglobin; Hp: haptoglobin; CD163: cluster of differentiation 163; CD91/LRP: cluster of differentiation/ low-density lipoprotein receptor related protein; *HRG*: heme responsive gene.

**Figure 3 ijms-21-05529-f003:**
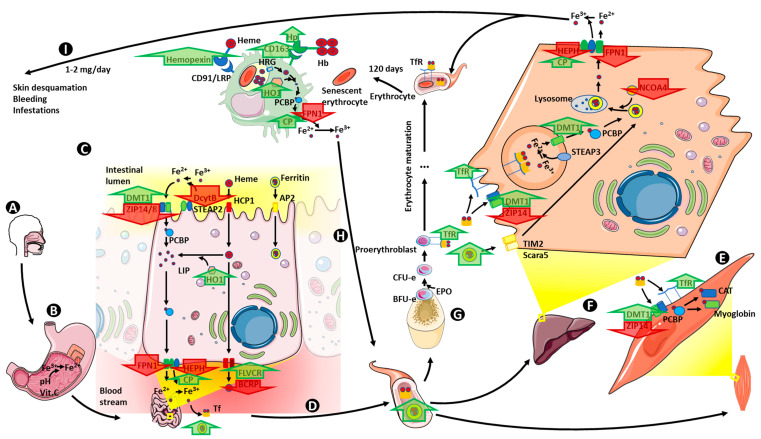
Iron metabolism dysregulation in obesity. Obesity influences iron metabolism at many steps of the cycle. No alterations have been proven in iron ingested from the diet (**A**), nor in its reduction from Fe^3+^ to Fe^2+^ in the stomach (**B**). In the duodenum enterocytes in obese patients, DMT1 density increases at the apical membrane, whereas ZIP14 and 8 co transporters decrease. HO1 levels increase intracellularly and at the basal membrane CP and FLVCR increase, whereas FPN1, BCRP and HEPH decrease (**C**). Ferritin circulating levels increase (**D**). Myocyte and hepatocyte transferrin receptor density increases, together with an increase in DMT1 and a decrease in ZIP14 (**E**,**F**). Additionally, in hepatocytes, NCOA4 decreases, preventing ferritin degradation, and HEPH and FPN1 decrease, reducing the Fe^2+^ sent to circulation (**F**). Decreased levels of transferrin receptors are found in proerythroblasts (**G**). Changes in macrophages recycling Fe^2+^ from hemoglobin include an increase in hemopexin, Hp and CD163, indicating an increase in the capacity to uptake heme groups and hemoglobin, and an increase in CP with a decrease in FPN1, suggesting a decrease in the Fe^2+^ liberated to circulation, but an increased oxidizing capacity to Fe^3+^ (**H**). Finally, no changes in daily iron loss has been described (**I**). Vit. C: vitamin C; DMT1: divalent metal transporter 1; ZIP 14/8: Zrt–Irt-like protein 14 and 8; DcytB: duodenal cytochrome B; STEAP 2: six-transmembrane epithelial antigen of the prostate 2; HCP1: heme carrier protein 1; AP2: adaptor-related 2 protein; PCBP: poly (rC) binding protein; LIP: labile iron pool; HO1: heme oxygenase 1; FPN1: ferroportin 1; HEPH: hephaestin; CP: ceruloplasmin; FLVCR: feline leukemia virus subgroup C; BCRP: breast cancer-resistant protein; Tf: transferrin; TfR: transferrin receptor; CAT: catalase; Scara 5: scavenger receptor class A, member 5; TIM2: T-cell immunoglobulin and mucin domain-containing protein 2; STEAP 3: six-transmembrane epithelial antigen of the prostate 3; NCOA4: nuclear receptor coactivator 4; BFU-e: burst forming unit-erythroid; CFU-e: colony forming unit-erythroid; EPO: erythropoietin; Hb: hemoglobin; Hp: haptoglobin; CD163: cluster of differentiation 163; CD91/LRP: cluster of differentiation/ low-density lipoprotein receptor related protein; *HRG*: heme responsive gene. Upregulated transporters and enzymes are shown with a green arrow pointing upwards, whereas downregulated transporters and enzymes are shown with a red arrow pointing down.

**Table 1 ijms-21-05529-t001:** Iron metabolism disturbances described in obesity.

Iron Metabolism Level	Protein	Experimental Procedure	Result	Reference
Iron absorption	HO-1	Mice Ho-1 depletion	IR and inflammation	[108]
		Quantification in children	Higher levels in obesity	[109]
		Induction in HFD-mice and cell cultures	anti-inflammatory phenotype, insulin sensitivity, repressed adipogenesis	[110,111]
	BCRP	Obese humans	Intestinal dysfunction of the transporter	[112]
	FLVCR1	Quantification in HFD mice	mRNA levels positive relation with fasting glucose and negative with insulin resistance	[113]
	Dcytb	Quantification in HFD mice	Lower mRNA levels	[114]
	Hephaestatin	Quantification in HFD mice	Lower mRNA levels	[114]
	TfR1	Quantification in HFD mice	Higher mRNA and protein levels	[114]
	DMT1	Quantification in HFD mice	Higher mRNA and protein levels	[114]
	NHE3	Quantification in HFD mice	Reduced renal expression	[115]
	Zinc transporters	Obese women	Reduced mRNA levels	[116]
Iron storage	Ferritin	H-Ferritin deletion in HFD mice	Anti-obesogenic state	[119]
		Obese children	Relation with metabolically unhealthy obesity	[120]
Iron transport and utilization	FPN1	Quantification in HFD mice	Decreased levels of FPN1	[144,145]
		Quantification in HFD mice	Decreased levels of FPN1 in obese children	[147]
	Hephaestatin	Quantification in HFD mice	Lower mRNA levels	[114]
	CP	Obese adults and children	Higher circulating CP levels	[121,122,123]
	NCOA4	Quantification in HFD mice	Impaired expression	[124]
	sTfR	Obese humans	Related to BMI in hyperferritinemia	[125]
	Lactoferrin	Obese humans	Inversely related to BMI and obesity	[127,128]
	LCN2	Obese humans	Increased levels	[129]
		LCN2 knockout mice	Obesogenic and anti-thermogenic activity	[130]
		Obese children	Decreased levels	[131]
	Megalin	Endothelial megalin-deficient mice	Protective role against HFD-induced obesity	[132]
	Frataxin	Frataxin deletion in mice	Impaired oxidative metabolism and higher predisposition to suffer from high-caloric diet-induced obesity	[133]
Iron recycling	Haptoglobin	Obese women	Relation with BMI, HOMA-IR, fasting insulin or blood glucose blood levels	[134,135]
		Obese adults and children	Positive association with obesity and allele related to haptoglobin levels	[137]
		Obese children	Allele related to oxidative stress in obesity	[138]
	CD163	Obese humans	Elevated expression and relation with HOMA-IR	[139]
		Obese children submitted to improved lifestyle	Changes in CD163 associated with better insulin sensitivity	[140]
	Hemopexin	Quantification in cell cultures	Higher mRNA levels in inflammatory states and relation of hemopexin and adipogenesis	[141,142]
		Quantification in humans	Variations according to metabolic disease status and triglyceride levels	[142]
	*LRP*	Obese children	SNPs and epigenetic modifications related to BMI	[143]
Iron homeostasis regulation	Furin	Obese humans	Furin deficiency related to obesity risk	[146]
	Hepcidin	Obese children	Increased levels	[147]
	HJV	Obese humans	Increased mRNA levels in adipose tissue	[149]
	BMP 2	Obese humans	Higher levels	[150]
	BMP 4	BMP4 deletion in mice	Obesogenic effects	[150]
	BMP 7	HFD mice treated with BMP 7	Anti-obesogenic and anti-inflammatory effects	[150]
	BMPR1A	Obese humans	Increased mRNA levels and SNPs involved	[151]
	Matriptase-2	Deficient matriptase-2 mice	Protection against obesity and its complications	[152]
	SMAD proteins	HFD mice overexpressing SMAD isoforms	SMAD 2, 3 and 4 improve obesity-related metabolic parameters and inflammation. SMAD7 has detrimental effects by regulating MPK38 activity	[153]
	EPO	Mice	Anti-obesogenic effects	[154,155,156]
	STAT3	Obese children and HFD mice	Lower mRNA levels	[157]
	HIF1	Mice	Obesogenic effects	[159,160]
	HIF1	Mice	Anti-obesogenic effects	[161]
	HIF2	Mice	Anti-obesogenic effects	[162]
	miRNA122	Obese humans	Increased levels	[163]

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
