# Peer review of "Iron Metabolism in Obesity and Metabolic Syndrome"

_ijms, 2020, doi:10.3390/ijms21155529_

Round 1
Reviewer 1 Report
This is a comprehensive review about iron metabolism in childhood obesity.
The paper is generally very well written and covers important facts.
The authors correctly state in the conclusions that both, oxidative stress and inflammation are related to obesity, and elsewhere, that oxidative stress is related to Fenton´s reaction.
Unfortunately, through the paper, this absolutely crucial distinction in iron redox state is not always, i.e. mostly not, clearly given.
Therefore, I strongly suggest that the authors screen their paper and specify where necessary and possible.
Some examples:
Lines 217 – 222: iron regulation is strictly dependent on redox state (Fe(II) / Fe(III), also transport by transferrin (Fe(III))
Line232: release free iron to LIP (should read iron ions, and supposedly as Fe(II)
Line 240: wrong expression “elements”: the authors mean here metabolites, not elements such as Cu, Zn or any other.
Line 259: in which 4500 iron ions can be stored: yes after oxidation, stored as Fe(III); removing redox-active Fe(II) from cellular chemistry. Please complete.
279: transferrin-bound iron (III)
383: regulated by iron (which state) and regulates iron levels (which state)?
Author Response
Request: Through the paper, the absolutely crucial distinction in iron redox state is not always, i.e. mostly not, clearly given. Therefore, I strongly suggest that the authors screen their paper and specify where necessary and possible.
Answer: Thank you very much for raising this point. We agree with the reviewer that this distinction should always be made. Hence, we proceeded to screen the paper as suggested, changing as necessary.
The iron redox state has been specified all through the text. Changes clearly highlighted, using the "Track Changes" function in Microsoft Word
Reviewer 2 Report
This is a review article regarding to the childhood obesity and its association with the development of several comorbidities such as nonalcoholic fatty liver disease, dyslipidemia, type 2 Diabetes Mellitus, non-congenital cardiovascular disease, chronic inflammation and anemia among others. Within this tangled interplay between these comorbidities and associated
pathological conditions, obesity has been closely linked to important perturbations in iron metabolism.
This is a well organized manuscript and provides comprehensive discussion about the childhood obesity and metabolic disorders.
My suggestion is that authors may consider use more infographic method to present the result.
Author Response
Request: My suggestion is that authors may consider use more infographic method to present the result
Answer: Thank you very much. Indeed we all agree with the suggestion, and have accordingly introduced two figures. The first new figure (manuscript’s fig. 2), summarizes physiological iron metabolism, since its intake through diet, to the elimination by bleeding, desquamation and infestations, going through absorption, use, storage and recycling. The second new figure (i.e. fig.3), summarizes the changes described to date in each of these processes in patients with obesity.
We hope it will satisfy the reviewer’s request for infography.